# Fuel Cell Application for Investigating the Quality of Electricity from Ship Hybrid Power Sources

Hyeonmin Jeon [1], Seongwan Kim [1] and Kyoungkuk Yoon [2,*]

1   Department of Marine System Engineering, Korea Maritime and Ocean University, 727 Taejong -ro, Busan 49112, Korea
2   Department of Electrical Engineering, Colleges of Korea Polytechnic, 155 Sanjeon-ro, Ulsan 44482, Korea
*   Correspondence: kkyoon70@kopo.ac.kr; Tel.: +82-105-541-0424

**Abstract:** Since recent marine application of fuel cell systems has been due largely limited to small-sized ships, this paper was aimed to investigate the technical applicability of molten carbonate fuel cell (MCFC) for medium and large-sized ships, using a 180 kW class hybrid test bed with combined power sources: A 100 kW MCFC, a 30 kW battery and a 50 kW diesel generator. This study focused primarily on determining whether the combined system designed in consideration of actual marine power system configuration could function properly. A case study was conducted with a 5500 Twenty-foot Equivalent Unit (TEU) container vessel. The operation profile was collected and analyzed in order to develop electric load scenarios applicable to the power system. Throughout the experiment, we evaluated the power quality of the voltage and frequency in the process of synchronization and de-synchronization across the power sources. Therefore, research results revealed that power quality continued to be excellent. This outcome provides insight into the technical reliability of MCFC application on large marine vessels.

**Keywords:** Molten carbonate fuel cell (MCFC); Hybrid test bed; Operation profile; Power quality

## 1. Introduction

Recently, hydrogen has started to be regarded as an alternative ocean fuel source. As a result, studies on hydrogen-fueled fuel cells have been actively conducted in Europe and the United States [1–3].

Fuel cells are known as a technology in which the chemical energy associated with hydrogen molecules is converted to electricity and thermal energy through electrochemical reactions with air. Thanks to the minimization of emissions, the noise in operation and high adaptability of various fuel sources, fuel cells are considered a next-generation technology for clean production [4,5].

In particular, fuel cell-based power plants are expected to reduce greenhouse gas emissions by 30% compared to fossil fuel-based power generation. Considering such benefits, many developed countries have been carried out a number of projects to stimulate the fuel cell application to industries [4–10].

According to a report of the European Maritime Safety Agency (EMSA), since the first project 'US SSFC' was launched in 2000, 24 projects have been made available to facilitate the application of marine fuel cells. Eleven of them selected the Proton Exchange Membrane Fuel Cell (PEMFC) type composed of polymer resin with several advantages: Relatively low operating temperature, about 80 °C; shorter time to reach the operating temperature; unnecessary of peripheral devices [11].

On the other hand, to ensure the reliability of the stakes, it requires pure hydrogen which is operated in a very low temperature with the late response time. In order to reduce such drawbacks, expensive catalyst and electrode are applied [12–14].

To obtain pure hydrogen, a separate reforming system is additionally required. That limits the application of PEMFC for the propulsion of medium-large ships. The Zero Emissions Ships (ZEMShip)

project has demonstrated the excellent performance of a fuel cell by applying a 96 kW fuel cell to the Motor Vessel (MV) 'Alsterwasser' [15–19].

In 'FellowSHIP' and 'Molten Carbonate Fuel Cells for Waterborne Application (MC-WAP)' projects, molten carbonate fuel cell (MCFC) type was applied to supplement auxiliary power rather than the main propulsion power. In particular, 'FellowSHIP' from 2003 to 2017, applied MCFC fuel cells for MV 'Viking Lady', 6,000 Deadweight Tonnage (DWT) Offshore Support Vessel (OSV), The project results showed excellent performance of the fuel cell in reducing emission levels [8,20–22].

Because MCFCs can operate at high temperatures, low-cost catalysts are available, which simplifying system design and reducing costs. In addition, even with long voyages, this type of fuel cell can utilize natural gas or coal gas as a direct fuel instead of using an external reformer. These advantages may be suitable for application as a major source of power for the ship's fundamental loads. Despite many research and projects for fuel cell applications in the marine industry, attempts to use the MCFC type for medium and large vessels for propulsion are scarce. Given this background, this study was motivated to investigate the suitability of the MCFC for the large vessel propulsion [23–27].

This paper was focused on analyzing the power quality of each power source in synchronization and breakaway. A test bed with a capacity of 180 kW was constructed using 100 kW fuel cells, 30 kW batteries and 50 kW diesel generator.

An actual voyage data for 5,500 TEU container vessel was used to verify the power quality of the fuel cell. Three load scenarios were developed by examining the performance characteristics of the fuel cell, battery and diesel generator systems based on normal navigational conditions. Each developed scenario was applied to the hybrid power sources, and the quality of voltage and frequency was examined during synchronization and breakaway phases.

## 2. Methodology

Given that the past research was largely focused on the marine application of MCFC as the main power source for small-sized vessel, this research would be a record of the first research for investigating the practicability of MCFC for medium- and large-sized vessels. To achieve this goal, a 5,500 TEU class container ship was selected as a case ship, and its operating data was applied for the test bed simulation in order to analyze the power quality of the system.

For this research, a combined power source test bed composed of MCFC, battery and diesel engine was constructed. Based on the developed load scenario from the case ship, we analyzed the quality of voltage and frequency in response to the synchronization and de-synchronization between each power source in the test bed and evaluated whether the results could meet the current regulations. Figure 1 shows the outline of the research process.

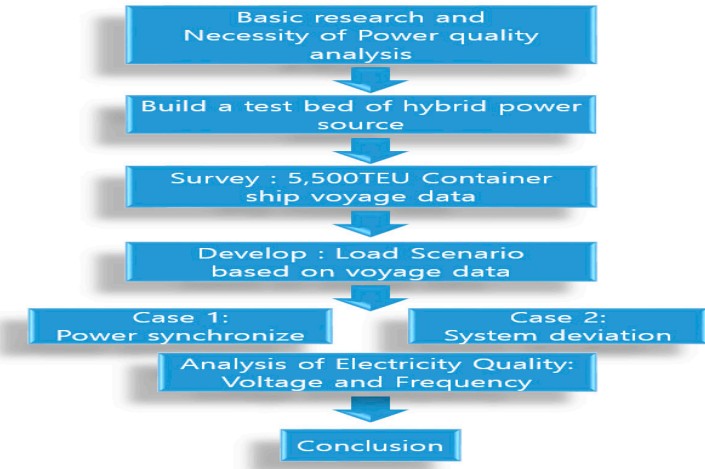

**Figure 1.** Flowchart for research procedure.

### 3. Design and Construction of Fuel Cell-Based Hybrid Power Source Test Bed

*3.1. Design*

Figure 2 shows the basic structure of the energy management system (EMS) used to control the hybrid power system in the test bed in accordance with load variations. In detail, the main parts of the test bed consist of MCFC system, energy storage system (ESS), diesel generator, load bank and EMS and electric power switching system (EPSS). In addition, the test bed was also optimally programmed to reliably synchronize hybrid power systems.

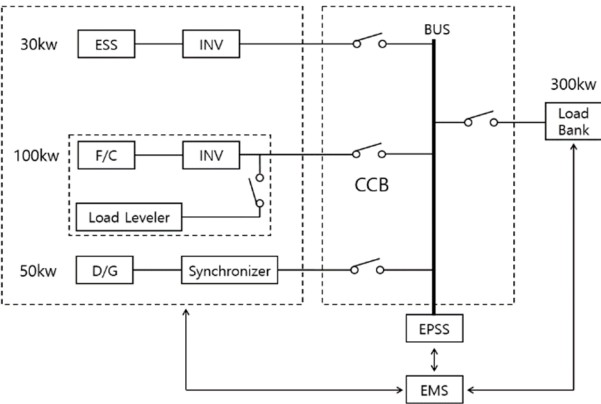

**Figure 2.** Basic outline of hybrid power systems for the test bed.

*3.2. Components of the Test Bed for Fuel Cell-Based Ship Hybrid Power Systems*

Figure 3 shows the arrangement of the fuel cell-based ship hybrid power system test bed.

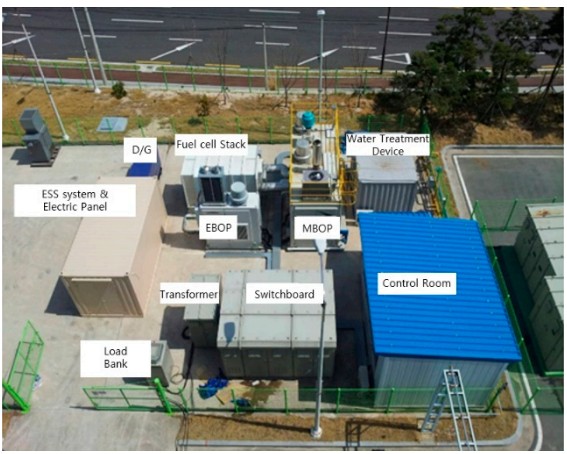

**Figure 3.** Actual placement of the test bed.

3.2.1. MCFC System

The fuel cell to be applied to the test bed is a 300 kW class MCFC system 'DFC300MA' model manufactured by 'POSCO Engineering' [28]. It is also a model that 'FuelCell Energy (FCE)' in the USA has developed as a basic model. Various development and experiments in several stages made the most optimized system at present. The fuel cell system consists of a stack module. EBOP (electric balance of plant) and MBOP (machinery balance of plant). The specification of the fuel cell system is shown in Table 1.

**Table 1.** Specification of the fuel cell system.

| Specification for 'DFC300MA'. | |
| --- | --- |
| Power Output | - |
| -Rated output | 250 kW |
| -Voltage | 380 - 480 VAC |
| Frequency | 50 ~ 60 Hz |
| -Power Quality | Per IEEE 519 |
| Emissions | - |
| -NOx | <0.02 lb/MWh |
| -SOx | <0.001 lb/MWh |
| -CO | <0.05 lb/MWh |

Figure 4 shows the peripheral equipment for the fuel cell system: Particularly, a fuel injection part for natural gas supply, a water injection part for making ultrapure water, a water discharge part and nitrogen/mixed gas injection part for stack protection. The Air injection part and the water discharge part are located on the upper side of MBOP within which two sets of exhaust fans are fitted.

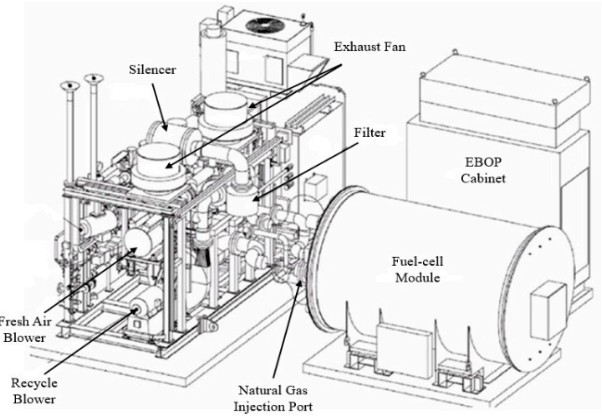

**Figure 4.** Overall configuration for fuel cell system.

Figure 5 describes the concept of the electricity generation process. The system consists of three modes: A 'heat-up mode' for increasing the initial temperature of the fuel cell stack module, a 'ramp-up mode' for increasing the power output to the rated output, and an 'operating mode' for continuing the rated output.

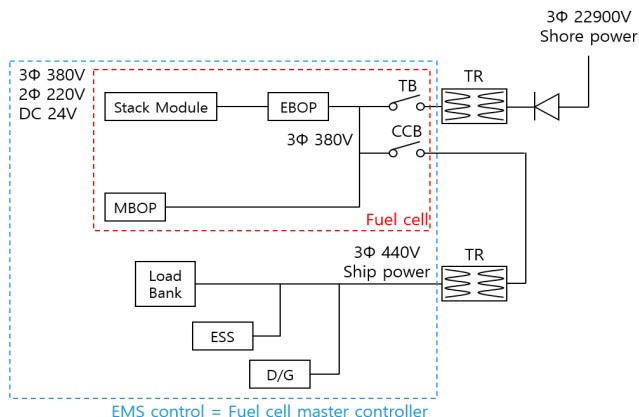

**Figure 5.** Concept of the fuel cell power generation.

In the 'heat-up' mode of the fuel cell system, as shown in Figure 5, the Terminal Breaker (TB) is closed, and the fuel cell is operated by receiving electricity from the system. Since only the fuel cell consumes power, the Customer Critical Breaker (CCB) is opened. Respectively, it is designed to close the CCB to charge the ESS internal battery. In the 'ramp-up' mode (when the 'heat-up' is completed, and the fuel cell outputs power), the CCB is closed, and the EMS judges whether the ESS is on or off.

### 3.2.2. Energy Storage System (ESS)

ESS refers to a small/medium-sized electrical storage facility that is to store electrical energy and use it when necessary in aids of a distributed power source in a micro-grid. As shown in Figure 6, ESS comprises the battery and power conditioning system (PCS).

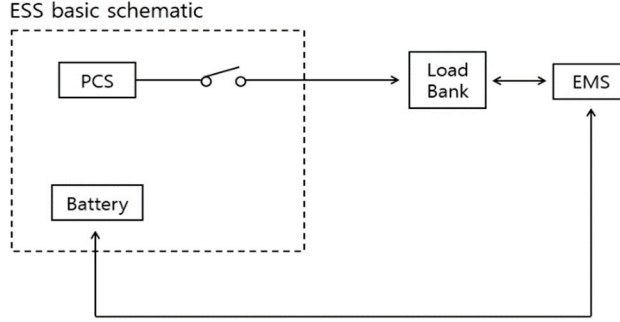

**Figure 6.** Configuration diagram energy storage system (ESS).

A lead-acid type battery was applied to the test bed. According to Table 2, it was modelled based on the two-way grid connected type.

**Table 2.** Specification of ESS.

| ESS Specification | |
|---|---|
| Input Voltage | 440VAC 3phase 3wire(60 Hz) |
| Output Voltage | 440VAC 3phase 3wire(60 Hz) |
| Load Capacity | 30 kW |
| Battery Ampere-hour | 400 Ah |
| Inverter Type | Bi-directional grid connected type |

The PCS is designed to perform bidirectional power control for DC power and AC power between the grid and the rechargeable battery, to improve the reliability of the power system, and to supply the stored energy quickly at peak power demand.

Therefore, The PCS is to balance active powers from hybrid power systems when the load fluctuation, accidental power supply, or load drop occurs by means of charging or discharging the battery and to contribute to enhancing system stabilization by adjusting the frequency. In addition, it has a functionality to monitor the state of charge (SOC) of the battery in real time and to control the temperature, current and voltage so that it can make the system to be operated with high reliability. It also provides surge protection, automatic overcharge / overload protection as well as overvoltage protection [29,30].

### 3.2.3. Diesel Generator System

A 50 kW diesel generator used in the test bed is a revolving-field type using a permanent magnet. This system is soundproofed to 75 db or less. As shown in Figure 7, and Tables 3 and 4, it has the synchronous speed of 1800 rpm with four poles.

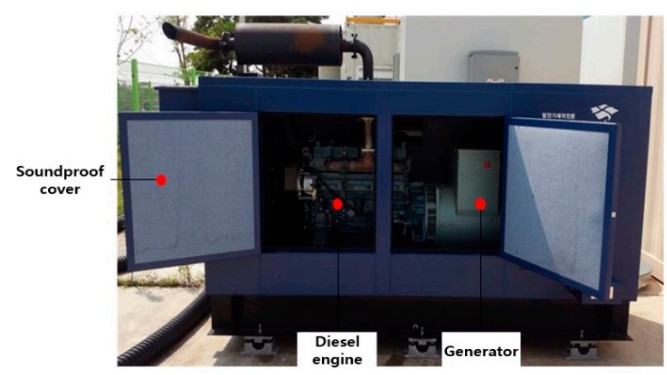

**Figure 7.** Appearance of diesel generator.

**Table 3.** Major specification of diesel generator engine.

| Item | Specification |
|---|---|
| Standby Power Rating | >95 PS |
| Engine Type | 4 stroke, water cooled |
| Revolution | 1800 RPM |
| Number of Cylinders | 6 |
| Fuel | Diesel |

**Table 4.** Major specification of diesel generator.

| Item | Specification |
|---|---|
| Type | Salient Pole Generator |
| Standby Power Rating | 50 kW / 62.5 kVA |
| Prime Power Rating | 45 kW / 56 kVA |
| Voltage | 440 / 254 V |
| Current | 82 A |
| Phase and Wire | 3phase 4wire |
| Frequency | 60 Hz |
| Number of Pole | 4 P |
| Revolution | 1800 RPM |
| Excitation System | Blushless Self-Exciter |

### 3.2.4. Load Bank

The load bank is a device designed to provide an electrical load for testing power sources, such as generators or uninterruptible power sources. In the case of a load bank used in a test bed, a power line is constructed for load testing, and the current load factor is calculated in the EMS and adjusts the load output from the load bank. The consumption of the voltage and current during this operation was monitored by an analog signal.

As shown in Table 5, the 300 kW load bank adopted the second type of iron chrome (FCHW-2) with high resistivity and low resistance against temperature increase and used a forced air-cooled load bank which was connected in parallel so that the load capacity could be adjusted.

**Table 5.** Specification of the load bank.

| Rated Power | 300 Kw | | | | |
|---|---|---|---|---|---|
| **Rated Voltage** | 3∅ 440 V, 60 Hz | | | | |
| **Current and resistance value for each capacity** | | | | | |
| Unit Capacity (kW) | Quantity | Total Capacity (kW) | Unit Current (A) | Unit Resistance | Cabinet |
| 0.1 | 1 | 0.1 | 0.13 | 1936Ω*3∅ | |
| 0.2 | 1 | 0.2 | 0.26 | 968Ω*3∅ | |
| 0.4 | 1 | 0.4 | 0.52 | 484Ω*3∅ | |
| 0.8 | 1 | 0.8 | 1.04 | 242Ω*3∅ | |
| 1 | 1 | 1 | 1.31 | 193.6Ω*3∅ | |
| 2 | 1 | 2 | 2.62 | 96.8Ω*3∅ | 1 box |
| 4 | 1 | 4 | 5.24 | 48.4Ω*3∅ | |
| 8 | 1 | 8 | 10.49 | 24.2Ω*3∅ | |
| 16 | 1 | 16 | 20.99 | 12.1Ω*3∅ | |
| 32 | 1 | 32 | 41.98 | 6.05Ω*3∅ | |
| 60 | 4 | 240 | 78.78 | 3.22Ω*3∅ | |
| Total | 15 | 304.5 | 399.7 | - | |

### 3.2.5. Energy Management System

EMS is a control system configured to monitor the voltage, current, output amount, and system status of each device in real time to operate the system stable. It is to balance the load of the load bank in real time according to the load variations so that each device can be synchronized properly [31–35].

The EMS and each power source—namely MCFC, diesel generator and ESS—are configured to send and receive status and operation commands of the device via the interface as presented in Figure 8.

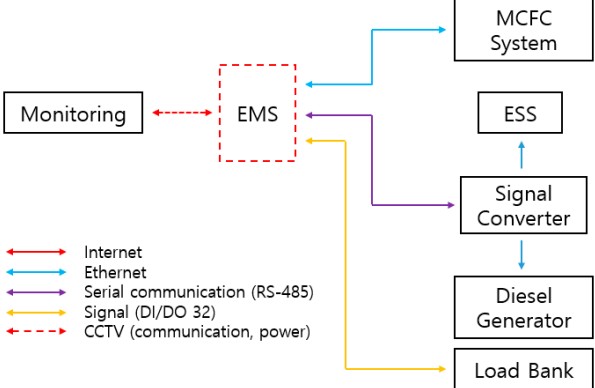

**Figure 8.** Integrated power control system configuration.

The power source (fuel cell, battery, diesel generator) constituting the hybrid system was controlled by EMS, according to the output range. In the output range of 0−100 kW, the fuel cell produces an output of 100 kW as the base load. When the required load is less than 100 kW, the battery is set to be charged, or the energy is sent to load leveler.

In addition, in the 100−130 kW output period, the battery power is additionally supplied to the load with fuel cell power, and the diesel generator is controlled to operate in the output period of 130 kW or more. When the proposed hybrid system was operated on the basis of the above conditions, the power quality was analyzed by measuring the voltage variation rate and frequency variation rate for each scenario.

## 4. System Composition for Power Quality Analysis

Figure 9 represents the configuration of MCFC system so that we can monitor and analyses frequency stabilization time as well as the voltage fluctuation rate and frequency fluctuation rate which can occur in the process of synchronizing the other power sources, such as ESS and diesel generator.

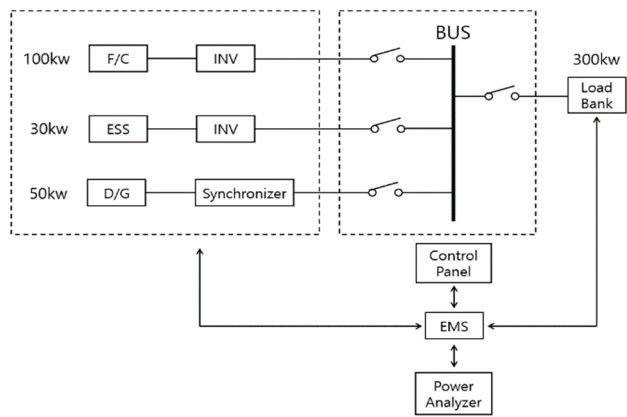

**Figure 9.** Overall system configuration for power quality analysis.

As shown in the figure, the overall system is designed for power quality analysis which is to measure the voltage, current, power, frequency, and power factor generated during power conversion through the synchronous test, thereby to evaluate the suitability of the combined power source to a marine vessel.

Figure 10 shows the monitoring system configuration for power quality analysis. We could monitor power data (voltage, current, frequency, voltage variation, frequency variation, apparent/active/reactive power and power factor) in real time and store them on the LabVIEW interface.

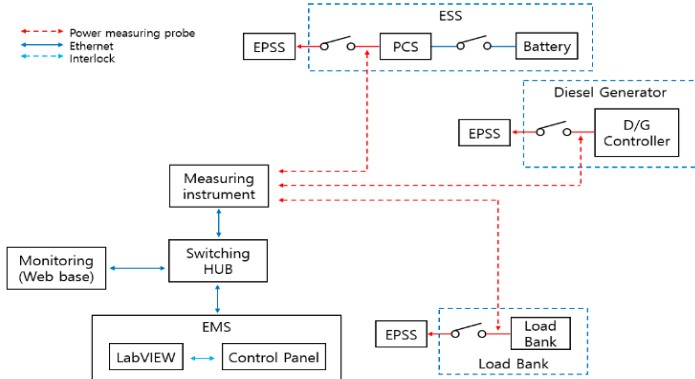

**Figure 10.** Monitoring system configuration for power quality analysis.

Twelve channels of the measuring points were assigned on MCFC, ESS and secondary power supply R, T of diesel generator. A high-voltage probe (TP0200) was used for the measurement as pictured in Figure 11, whereas a current measuring clamp i410 type was applied, as shown in Figure 12.

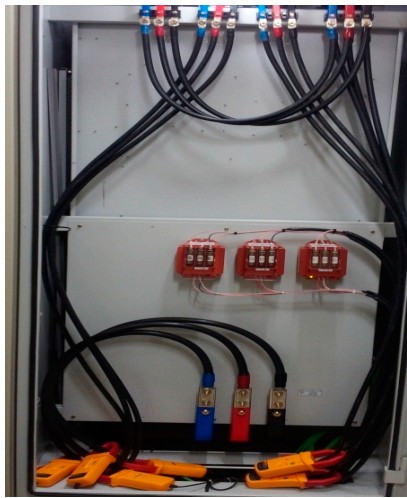

**Figure 11.** Installed voltage measuring equipment.

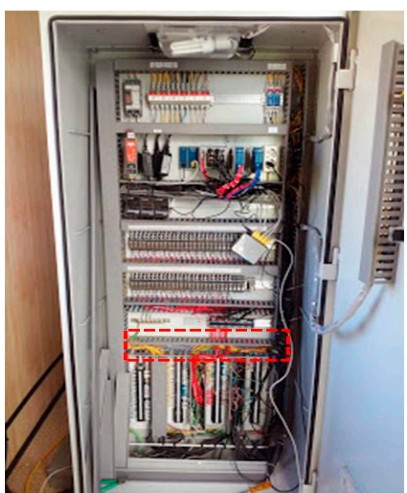

**Figure 12.** Installed current measuring equipment.

## 5. Development of Load Scenario for Power Quality Analysis of Fuel Cell-Based Marine Hybrid Power System

### 5.1. Load Scenario Development

In this study, when the MCFC, which is responsible for the base load of the fuel cell-based combined power source, is applied to the ship with other power sources, such as ESS and diesel generator, we developed a load scenario for power quality analysis through voltage fluctuation rate, frequency fluctuation rate, and frequency stabilization time for power faults that may occur in the process of synchronizing to or disconnection from a single bus line. Scenario requested sailing information of the 5,500 TEU vessel of 'H' shipping company and collected and analyzed the data.

Based on the data of main engine and generator output of 5,500 TEU class container ship, we analyzed the load variation characteristics in normal seagoing operation in order to investigate whether each power source can maintain proper power quality for load fluctuation, such as synchronization and system deviation. We have developed a Load Scenario to determine if it is possible.

Using the real-time operation data, the load case was developed based on the following steps.

- Analyzes the load pattern according to the size of the fluctuating load when analyzing operational data.

- Exclude generator load which is 2.5–5% of the main engine loads as documented in IMO MEPC.1 / Circ.681
- Develop Load Scenario for verifying load followability of the fuel cell-based ship's hybrid power systems according to load variation
- For cargo ships with the main engine power of 10000 kW or above, $P_{AE}$ is defined as Equation (1);

$$P_{AE(MCRME>10000kW)} = (0.025 \ X \ \sum_{i=1}^{nME} MCR_{MEi}) + 250. \tag{1}$$

- For cargo ships with the main engine power below 10000 kW, $P_{AE}$ is defined as equation (2);

$$P_{AE(MCRME<10000kW)} = (0.05 \ X \ \sum_{i=1}^{nME} MCR_{MEi}). \tag{2}$$

### 5.2. Development of Load Scenario Using Real-Time Operation Load Pattern

#### 5.2.1. Sync Phase and Conditions for Normal Seagoing Case 1

Figure 13 and Table 6 summarize the synchronization and off-gird of the load pattern of the normal seagoing case 1 with the peak load of 127 kW of the combined power source. ESS is configured to discharge after one synchronization at a time when the total load of the complex power source is over 100 kW, and then it is changed to the charging mode by the ESS SOC reference value, and the charging is progressed in the state connected to the system.

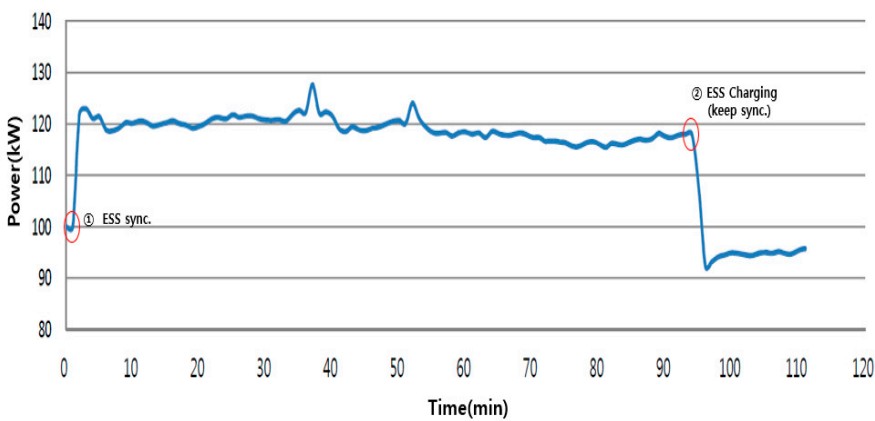

**Figure 13.** Synchronization and system deviation for load pattern of normal seagoing case 1.

**Table 6.** Synchronization condition in normal seagoing case 1.

| Status | Power Sources |
|---|---|
| ① ESS sync. | MCFC + ESS |
| ② ESS Charging (keep sync.) | MCFC + ESS |

#### 5.2.2. Sync Phase and Conditions for Normal Seagoing Case 2

Figure 14 and Table 7 summarize the synchronization and out of gird areas when the load of the combined power source reaches 140 kW according to the load pattern of the normal seagoing case 2. The ESS is synchronized with the MCFC at the moment when the total load exceeds 100 kW. The diesel generator is synchronized with the system at a total load of 130 kW or more, and is configured so that when the load drops below 130 kW, the system disengages. When the load drops below 100 Kw, the

ESS is also separated from the MCFC, and the scenario is configured to synchronize with the ESS as soon as the total load rises above 100 kW again.

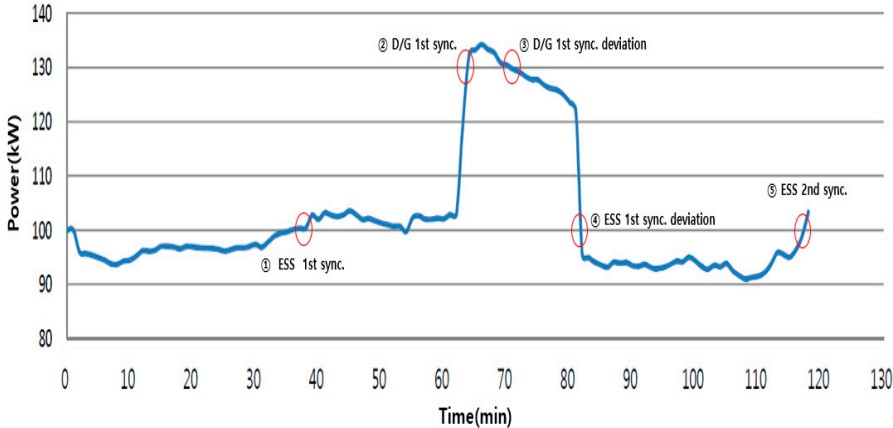

**Figure 14.** Synchronization and system deviation for load pattern of normal seagoing case 2.

**Table 7.** Synchronization condition in normal seagoing case 2.

|  | Status | Power Sources |
|---|---|---|
| ① | ESS $1^{st}$ sync. | MCFC + ESS |
| ② | D/G $1^{st}$ sync. | MCFC + ESS+ D/G |
| ③ | D/G $1^{st}$ sync. deviation | MCFC + ESS |
| ④ | ESS $1^{st}$ sync. deviation | MCFC |
| ⑤ | ESS $2^{nd}$ sync. | MCFC + ESS |

### 5.2.3. Sync Phase and Conditions for Normal Seagoing Case 3

Figure 15 and Table 8 summarize the synchronization and off grid part of the normal seagoing case 3. In the ESS, there are two synchronization periods and system outages repeatedly according to the load of the combined power source. In the section where the charge interval and the discharge interval are repeated, charging and discharging are performed in a state, in which the system synchronization is maintained. The diesel generator has two synchronization periods and a system outage period.

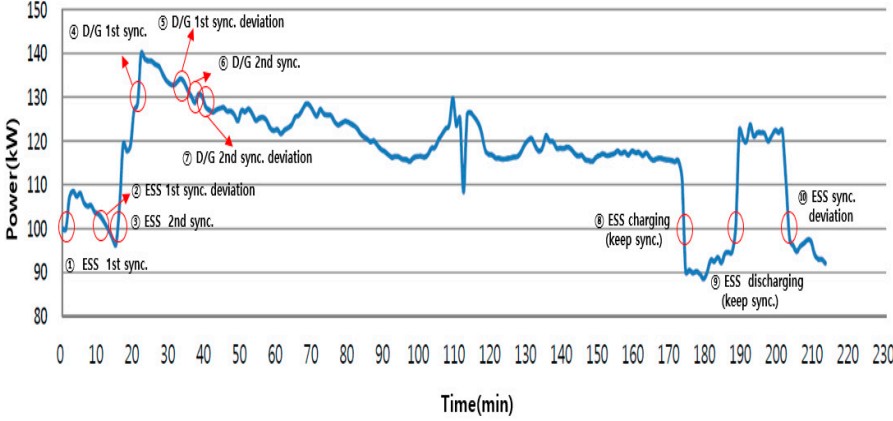

**Figure 15.** Synchronization and system deviation for load pattern of normal seagoing case 3.

**Table 8.** Synchronization condition in normal seagoing case 3.

| Status | Power Sources |
|---|---|
| ① ESS 1st sync. | MCFC + ESS |
| ③ ESS 1st sync. deviation | MCFC |
| ④ ESS 2nd sync. | MCFC + ESS |
| ⑤ D/G 1st sync. | MCFC + ESS + D/G |
| ⑥ D/G 1st sync. deviation | MCFC +ESS |
| ⑦ D/G 2nd sync. | MCFC + ESS + D/G |
| ⑧ D/G 2nd sync. Deviation | MCFC + ESS |
| ⑨ ESS Charging (keep sync.) | MCFC + ESS |
| ⑩ ESS Discharging (keep sync.) | MCFC + ESS |
| ⑪ ESS sync. Deviation | MCFC |

## 6. Procedures for Power Quality Test for Ship Hybrid Power System

*6.1. Synchronization Procedure between Hybrid Power Sources*

### 6.1.1. The procedure for ESS Synchronization

ESS system voltage is followed by priority, and when the system voltage is transmitted to the power conversion device (PCS) of the ESS via the sensor, the voltage is controlled by following the system voltage.

When the synchronous signal is transmitted from the EMS to the PCS, the PCS follows the system frequency. The PCS adjusts the frequency so that the system can be stably maintained. In this case, the PCS controls the system using the current and perform charging and discharging actions in accordance with the pre-setting power outputs.

The ESS synchronization time would be approximately 0.5 seconds during which PCS turns on, and EMS generates a synchronization signal in the normal operation state in which charging and discharging are enabled.

### 6.1.2. Procedure of Diesel Generator

When the synchronization signal is produced to the EMS in normal operation, the diesel generator adjusts the voltage based on the grid voltage and synchronizes with the system frequency. The synchronization controller of the diesel generator plays a role in maintaining the system stable by adjusting the frequency. At this time, the amount of electric power in the form of current control mode is limited, so that more current of electric power than the pre-set can be prevented from flowing to the system.

Once the EMS synchronization signal is generated in the normal operation state of the diesel generator, it takes about 3−4 seconds to synchronize to the system.

*6.2. Power Measurement Point and Selection Criteria When Synchronizing Hybrid Power Sources*

The effect of voltage and frequency on the system was investigated by measuring the synchronization time between the ship's combined power source in two times: Two seconds before synchronization and five seconds after synchronization. The measuring time was established based on Korean Classification Rule. Therefore, the built-in LabVIEW-based power analysis system for test bed experiment can store one power measurement data every 33.3 ms.

The ESS was configured to measure two seconds before synchronization and five seconds after synchronization in consideration of the short synchronization time; it could be confirmed through data that ESS took 7−10 seconds depending on the system condition when synchronizing.

The synchronization time of the diesel generator was designed to measure two seconds before synchronization and five seconds after synchronization in the same manner as ESS. However, in case

of diesel generator, it took about 10−12 seconds because the synchronization time usually took 3−5 seconds, which was confirmed through the results.

## 7. Analysis Result of Power Quality Data

### 7.1. Target of Voltage, Frequency Variation and Stabilization Time

With regard to synchronization of MCFC, ESS and diesel generation, the voltage, frequency variation, and frequency variation and stabilization time in the transient state of disconnection from the grid were tested with reference to the Korean Register of Shipping, "Class 6 Electrical Equipment and Control Systems" in Table 9 [36].

**Table 9.** Rules for the classification of electrical installation and control systems of the Korean Register of Shipping.

| Item | Regulation | | Stabilization Time |
| :---: | :---: | :---: | :---: |
| | **Steady State** | **Transient State** | |
| Frequency | ±5% | ±10% (5s) | <15s |
| Voltage | +6%, −10% | ±20% (1.5s) | - |

### 7.2. Test Results of Synchronization Interval and Breakaway Interval

#### 7.2.1. Test Results for ESS Synchronization and Breakaway

Figure 16 shows the voltage fluctuation trend of the ESS before and after the synchronization in the MCFC system. It can be found that the system voltage before the ESS synchronization encountered a slight deviation, but the follow-up control was effectively confirmed after the ESS synchronization.

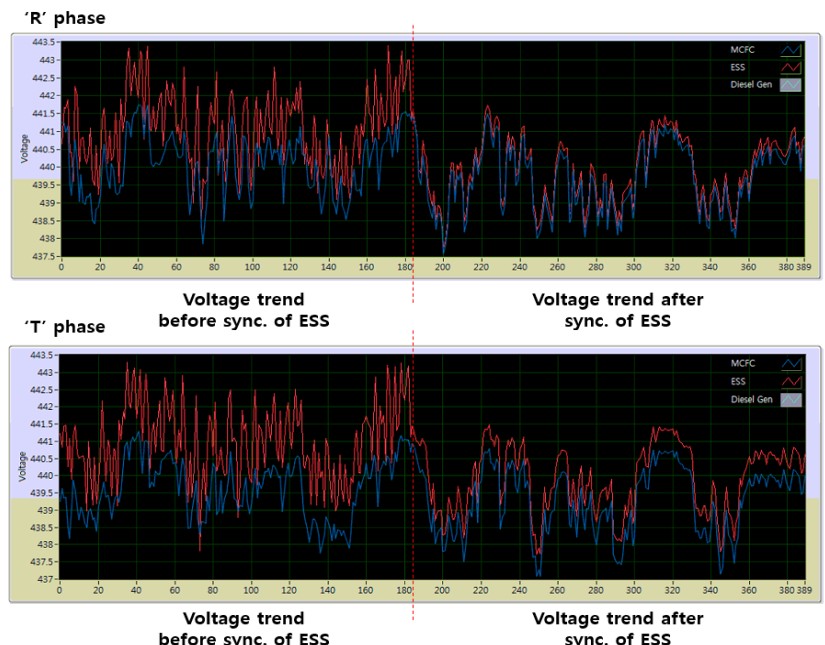

**Figure 16.** Voltage change during ESS synchronization.

In Figure 17, the synchronous signal from IMES followed the system voltage and frequency. The reference frequency for both MCFC and ESS was 60 HZ, and that was exactly matched each other after synchronization.

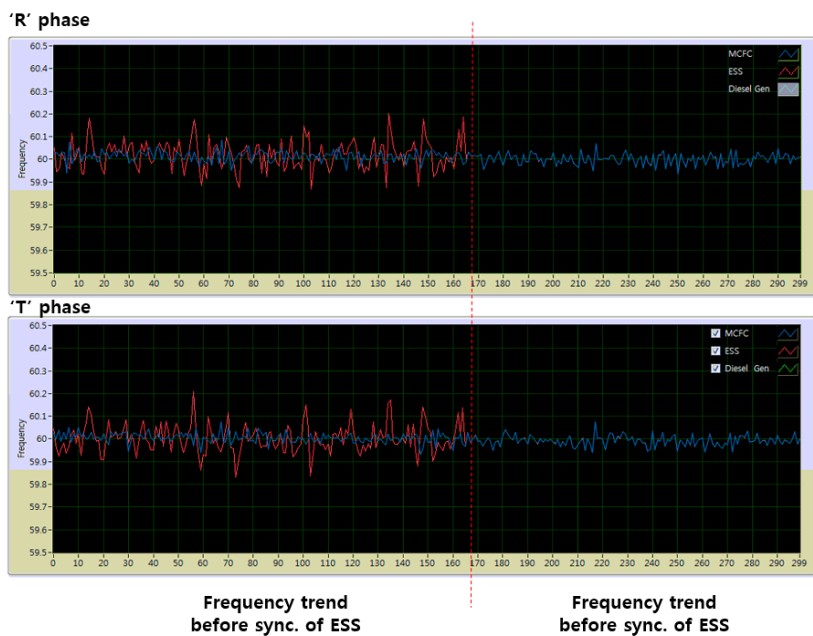

**Figure 17.** Frequency change during ESS synchronization.

When a system breakaway signal was generated from the IMES, the ESS was disconnected from the system. As shown in Figure 18, the separation time of the ESS from the system was estimated less than 0.5 seconds, which was the same as the synchronization time. The voltage fluctuation trend of the ESS, before and after synchronization in the MCFC system, was also clearly presented in the figure.

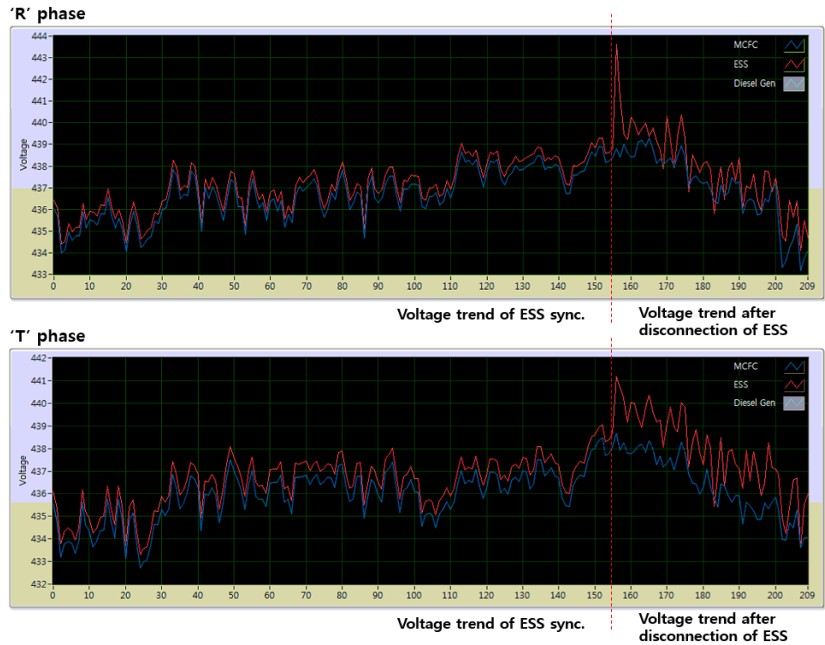

**Figure 18.** Voltage change when the ESS system was disconnected.

Figure 19 shows that the parallel operation of the MCFC and the ESS and the breakaway proceedings when the system breakaway signal was sent from the IMES to the ESS. In this case, the frequency was observed to be slightly hunt, but it did not affect the existing system.

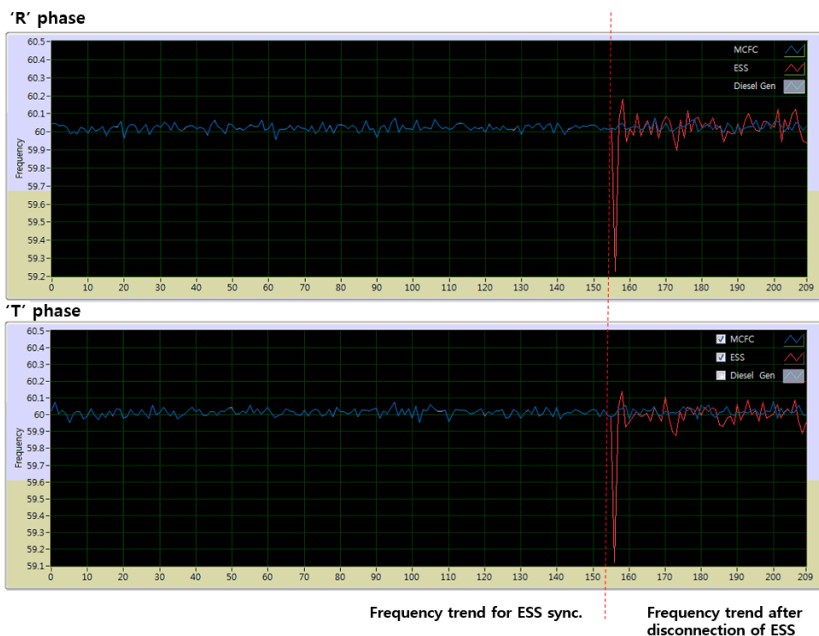

**Figure 19.** Frequency change in ESS system deviation.

### 7.2.2. Test Results For Diesel Generator Synchronization and Breakaway

Figure 20 shows the voltage fluctuation trend before and after the synchronization of the diesel generator with the MCFC and ESS systems. When the EMS sent the synchronization signal, it followed the system voltage so that the voltage could be synchronized—thereby, the system voltage was exactly matched.

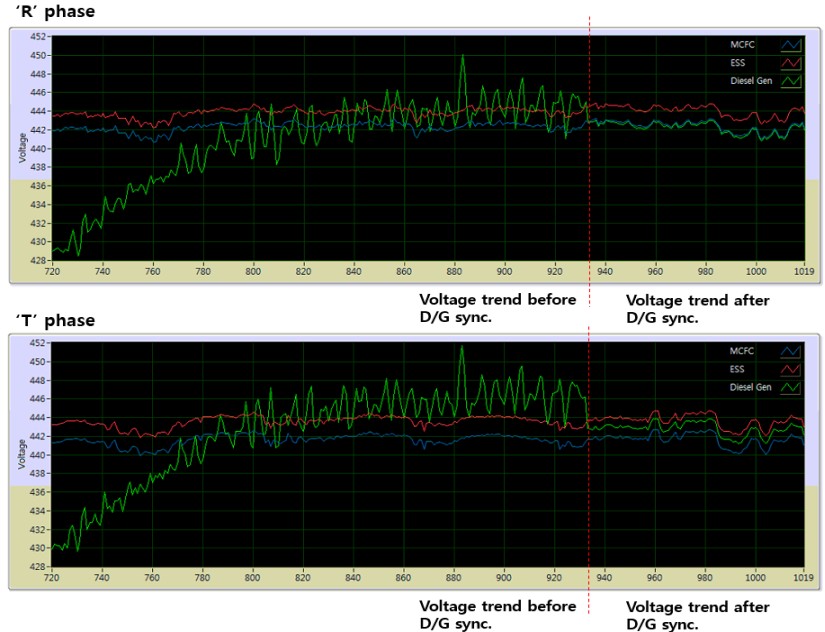

**Figure 20.** Voltage change when synchronizing diesel generators.

Figure 21 shows the frequency fluctuation trend before and after synchronizing the diesel generator with the MCFC and ESS systems. It revealed that once the EMS sent the diesel generator synchronization signal, it started to follow the system voltage and achieved the voltage synchronization. Thereafter, it followed the system frequency and attained frequency synchronization.

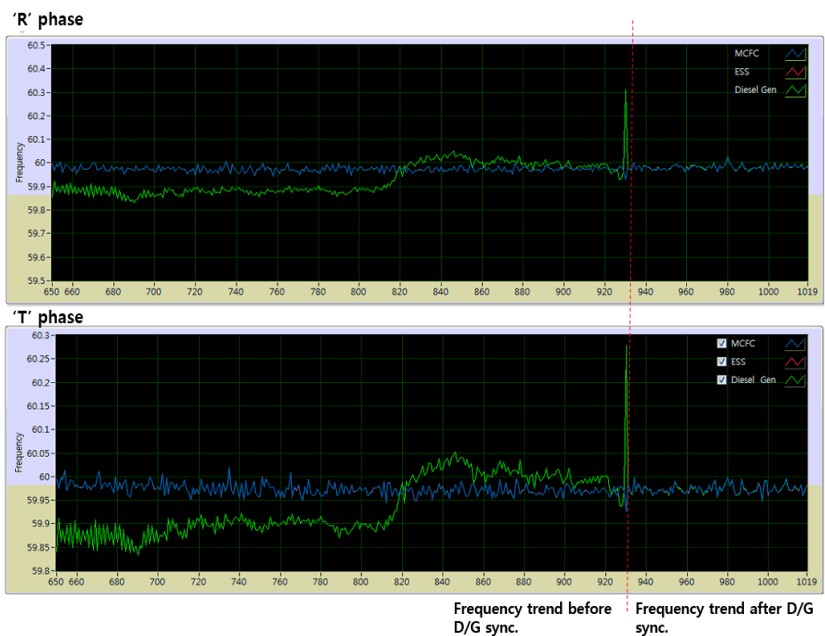

**Figure 21.** Frequency change when synchronizing diesel generators.

When the system separation signal was generated from the EMS, the diesel generator was disconnected from the system by the synchronization switch fitted in the synchronization controller. In Figure 22, a waveform similar to the synchronous voltage waveform was plotted in the same manner as in the system state before synchronizing the diesel generator.

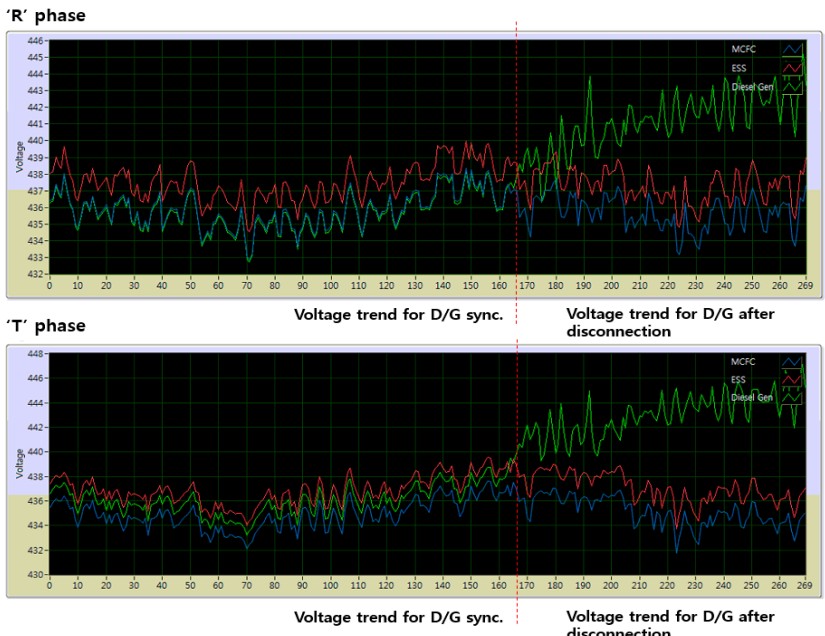

**Figure 22.** Voltage change when diesel generators are broken away.

Figure 23 shows the synchronous operation of all power systems in parallel. When the system breakaway signal went off to the IG-NT from the IMES, the frequency was observed to be a hunt, which occurred after the breakaway, thereby such an outage had no effect on the existing system.

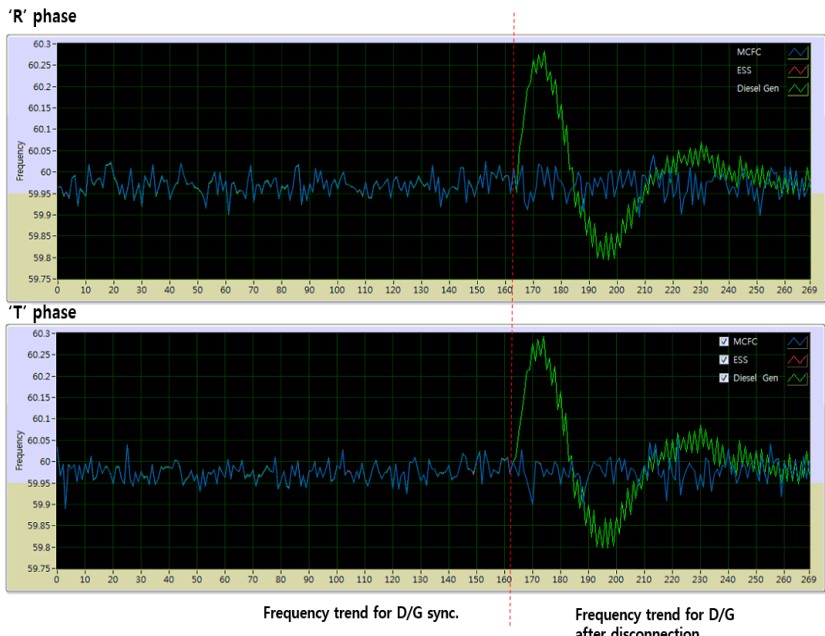

**Figure 23.** Frequency change when leaving diesel generator system.

*7.3. Analysis of Frequency Variation Rate and Frequency Stabilization Time Data Based on Load Scenarios*

7.3.1. Power Data before and after Synchronization for Normal Seagoing Case 1

Table 10 shows the analysis results pertinent to the voltage and frequency fluctuation measured in the load pattern of the normal seagoing case 1.

**Table 10.** Power data analysis before and after ESS synchronization.

| Item | Voltage Fluctuation | | | | | | Frequency Fluctuation | | | | | |
|---|---|---|---|---|---|---|---|---|---|---|---|---|
| | MCFC | | ESS | | D/G | | MCFC | | ESS | | D/G | |
| Condition of Synchronization | Bef | Aft | Bef | Aft | Bef | Aft | Bef | Aft | Bef | Aft | Bef | Aft |
| Maximum | 440.8 | 440.8 | 443.7 | 444.0 | - | - | 60.1 | 60.1 | 60.2 | 60.2 | - | - |
| Minimum | 438.4 | 438.3 | 440.1 | 438.5 | - | - | 60.0 | 59.9 | 59.8 | 59.9 | - | - |
| Average | 439.6 | 439.7 | 441.8 | 440.2 | - | - | 60.0 | 60.0 | 60.0 | 60.0 | - | - |
| Maximum rate of change (%) | 1.00 | 1.00 | 1.01 | 1.01 | - | - | 1.00 | 1.00 | 1.00 | 1.00 | - | - |
| Minimum rate of change (%) | 1.00 | 1.00 | 1.00 | 1.00 | - | - | 1.00 | 1.00 | 1.00 | 1.00 | - | - |
| Regulation (%) | ±1.00 | ±1.00 | ±1.00 | ±1.00 | - | - | ±1.00 | ±1.00 | ±1.00 | ±1.00 | - | - |

When the ESS was synchronized with the system, the voltage fluctuation rate was within ±1.01%, and the voltage fluctuation rate ranged from +6 to −10 %. Therefore, it could be confirmed that the voltage fluctuation hardly occurred during system synchronization, and the voltage characteristic was kept stable.

Likewise, the stability of frequency was also confirmed since the frequency variation rate was within ±1.01%, much lower than the standard of ±5%.

7.3.2. Power Data before and after Synchronization for Normal Seagoing Case 2

Tables 11–14 describe the analysis results from power data measured during synchronization and breakaway of the diesel generator under the normal seagoing case 2.

**Table 11.** Power data analysis before and after ESS synchronization.

| Item | Voltage Fluctuation | | | | | | Frequency Fluctuation | | | | | |
|---|---|---|---|---|---|---|---|---|---|---|---|---|
| | MCFC | | ESS | | D/G | | MCFC | | ESS | | D/G | |
| Condition of Synchronization | Bef | Aft | Bef | Aft | Bef | Aft | Bef | Aft | Bef | Aft | Bef | Aft |
| Maximum | 439.6 | 439.5 | 445.0 | 440.7 | - | - | 60.1 | 60.1 | 60.1 | 60.4 | - | - |
| Minimum | 437.2 | 437.5 | 440.8 | 437.8 | - | - | 60.0 | 60.0 | 0.0 | 60.0 | - | - |
| Average | 438.6 | 438.5 | 442.5 | 438.8 | - | - | 60.0 | 60.0 | 8.1 | 60.0 | - | - |
| Maximum rate of change (%) | 1.00 | 1.00 | 1.01 | 1.00 | - | - | 1.00 | 1.00 | 1.00 | 1.01 | - | - |
| Minimum rate of change (%) | 0.99 | 1.09 | 1.00 | 1.09 | - | - | 1.00 | 1.00 | 0.00 | 1.00 | - | - |
| Regulation (%) | ±1.00 | ±1.00 | ±1.01 | ±1.00 | - | - | ±1.00 | ±1.00 | ±0.13 | ±1.00 | - | - |

**Table 12.** Power data analysis before and after diesel generator synchronization.

| Item | Voltage Fluctuation | | | | | | Frequency Fluctuation | | | | | |
|---|---|---|---|---|---|---|---|---|---|---|---|---|
| | MCFC | | ESS | | D/G | | MCFC | | ESS | | D/G | |
| Condition of Synchronization | Bef | Aft | Bef | Aft | Bef | Aft | Bef | Aft | Bef | Aft | Bef | Aft |
| Maximum | 441.1 | 440.7 | 442.8 | 442.4 | 445.1 | 444.6 | 60.1 | 60.1 | 60.1 | 60.1 | 60.0 | 60.2 |
| Minimum | 438.0 | 437.9 | 439.6 | 439.6 | 426.9 | 438.2 | 60.0 | 60.0 | 60.0 | 60.0 | 59.8 | 59.9 |
| Average | 440.0 | 439.6 | 441.6 | 441.3 | 434.8 | 440.5 | 60.0 | 60.0 | 60.0 | 60.0 | 59.9 | 60.0 |
| Maximum rate of change (%) | 1.00 | 1.00 | 1.01 | 1.01 | 1.01 | 1.01 | 1.00 | 1.00 | 1.00 | 1.00 | 1.00 | 1.00 |
| Minimum rate of change (%) | 1.00 | 1.00 | 1.00 | 1.00 | 0.97 | 1.00 | 1.00 | 1.00 | 1.00 | 1.00 | 1.00 | 1.00 |
| Regulation (%) | ±1.00 | ±1.00 | ±1.00 | ±1.00 | ±0.99 | ±1.00 | ±1.00 | ±1.00 | ±1.00 | ±1.00 | ±1.00 | ±1.00 |

**Table 13.** Power data analysis before and after departing from the diesel generator system.

| Item | Voltage Fluctuation | | | | | | Frequency Fluctuation | | | | | |
|---|---|---|---|---|---|---|---|---|---|---|---|---|
| | MCFC | | ESS | | D/G | | MCFC | | ESS | | D/G | |
| Condition of Synchronization | Bef | Aft | Bef | Aft | Bef | Aft | Bef | Aft | Bef | Aft | Bef | Aft |
| Maximum | 438.5 | 439.2 | 440.2 | 440.9 | 438.5 | 444.4 | 60.1 | 60.1 | 60.1 | 60.1 | 60.1 | 60.4 |
| Minimum | 434.5 | 436.2 | 436.2 | 437.9 | 434.5 | 436.0 | 60.0 | 59.9 | 60.0 | 59.9 | 60.0 | 59.9 |
| Average | 437.2 | 437.6 | 438.9 | 439.3 | 437.2 | 439.6 | 60.0 | 60.0 | 60.0 | 60.0 | 60.0 | 60.0 |
| Maximum rate of change (%) | 1.00 | 1.00 | 1.00 | 1.00 | 1.00 | 1.01 | 1.00 | 1.00 | 1.00 | 1.00 | 1.00 | 1.01 |
| Minimum rate of change (%) | 0.99 | 0.99 | 0.99 | 1.00 | 0.99 | 0.99 | 1.00 | 1.00 | 1.00 | 1.00 | 1.00 | 1.00 |
| Regulation (%) | ±0.99 | ±0.99 | ±1.00 | ±1.00 | ±0.99 | ±1.00 | ±1.00 | ±1.00 | ±1.00 | ±1.00 | ±1.00 | ±1.00 |

**Table 14.** Power data analysis before and after ESS system outage.

| Item | Voltage Fluctuation | | | | | | Frequency Fluctuation | | | | | |
|---|---|---|---|---|---|---|---|---|---|---|---|---|
| | MCFC | | ESS | | D/G | | MCFC | | ESS | | D/G | |
| Condition of Synchronization | Bef | Aft | Bef | Aft | Bef | Aft | Bef | Aft | Bef | Aft | Bef | Aft |
| Maximum | 441.2 | 440.7 | 443.3 | 441.5 | - | - | 60.1 | 60.1 | 60.3 | 60.1 | - | - |
| Minimum | 438.1 | 437.2 | 438.4 | 438.8 | - | - | 60.0 | 60.0 | 60.0 | 59.9 | - | - |
| Average | 439.9 | 439.2 | 440.3 | 440.4 | - | - | 60.0 | 60.0 | 60.0 | 60.0 | - | - |
| Maximum rate of change (%) | 1.00 | 1.00 | 1.01 | 1.00 | - | - | 1.00 | 1.00 | 1.00 | 1.00 | - | - |
| Minimum rate of change (%) | 1.00 | 0.99 | 1.00 | 1.00 | - | - | 1.00 | 1.00 | 1.00 | 1.00 | - | - |
| Regulation (%) | ±1.00 | ±1.00 | ±1.00 | ±1.00 | - | - | ±1.00 | ±1.00 | ±1.00 | ±1.00 | - | - |

The voltage fluctuation rate was observed within ±1.01% when the ESS and the diesel generator were synchronized to the system. Since the voltage fluctuation was placed in the standard range of +6 to −10 %, the system stability was verified for this case as well.

The frequency deviation was within ±1.01%. In the same way, given the standard range of ±5%, frequency stability has been confirmed.

### 7.3.3. Power Data before and after Synchronization for Normal Seagoing Case 3

The analysis results for the voltage and frequency variation measured in the load pattern of normal seagoing case 3 are shown across Tables 15–18 which deal with power data for ESS synchronization and breakaway, as well as that for the diesel generator.

**Table 15.** Power data analysis before and after ESS synchronization.

| Item | Voltage Fluctuation | | | | | | Frequency Fluctuation | | | | | |
| | MCFC | | ESS | | D/G | | MCFC | | ESS | | D/G | |
| | Bef | Aft | Bef | Aft | Bef | Aft | Bef | Aft | Bef | Aft | Bef | Aft |
|---|---|---|---|---|---|---|---|---|---|---|---|---|
| Condition of Synchronization | | | | | | | | | | | | |
| Maximum | 442.9 | 442.6 | 445.2 | 442.9 | - | - | 60.1 | 60.1 | 60.2 | 60.1 | - | - |
| Minimum | 440.4 | 440.3 | 440.6 | 440.5 | - | - | 60.0 | 60.0 | 59.9 | 60.0 | - | - |
| Average | 441.7 | 441.5 | 442.7 | 441.8 | - | - | 60.0 | 60.0 | 60.0 | 60.0 | - | - |
| Maximum rate of change (%) | 1.01 | 1.01 | 1.01 | 1.01 | - | - | 1.00 | 1.00 | 1.00 | 1.00 | - | - |
| Minimum rate of change (%) | 1.00 | 1.00 | 1.00 | 1.00 | - | - | 1.00 | 1.00 | 1.00 | 1.00 | - | - |
| Regulation (%) | ±1.00 | ±1.00 | ±1.01 | ±1.00 | - | - | ±1.00 | ±1.00 | ±1.00 | ±1.00 | - | - |

**Table 16.** Power data analysis before and after ESS system outage.

| Item | Voltage Fluctuation | | | | | | Frequency Fluctuation | | | | | |
| | MCFC | | ESS | | D/G | | MCFC | | ESS | | D/G | |
| | Bef | Aft | Bef | Aft | Bef | Aft | Bef | Aft | Bef | Aft | Bef | Aft |
|---|---|---|---|---|---|---|---|---|---|---|---|---|
| Condition of Synchronization | | | | | | | | | | | | |
| Maximum | 441.3 | 441.4 | 441.6 | 443.5 | - | - | 60.0 | 60.0 | 60.0 | 61.3 | - | - |
| Minimum | 439.0 | 438.7 | 439.4 | 438.2 | - | - | 60.0 | 60.0 | 60.0 | 59.8 | - | - |
| Average | 440.4 | 440.3 | 440.8 | 440.8 | - | - | 60.0 | 60.0 | 60.0 | 60.0 | - | - |
| Maximum rate of change (%) | 1.00 | 1.00 | 1.00 | 1.01 | - | - | 1.00 | 1.00 | 1.00 | 1.02 | - | - |
| -Minimum rate of change (%) | 1.00 | 1.00 | 1.00 | 1.00 | - | - | 1.00 | 1.00 | 1.00 | 1.00 | - | - |
| Regulation (%) | ±1.00 | ±1.00 | ±1.00 | ±1.00 | - | - | ±1.00 | ±1.00 | ±1.00 | ±1.00 | - | - |

**Table 17.** Power data analysis before and after diesel generator synchronization.

| Item | Voltage Fluctuation | | | | | | Frequency Fluctuation | | | | | |
| | MCFC | | ESS | | D/G | | MCFC | | ESS | | D/G | |
| | Bef | Aft | Bef | Aft | Bef | Aft | Bef | Aft | Bef | Aft | Bef | Aft |
|---|---|---|---|---|---|---|---|---|---|---|---|---|
| Condition of Synchronization | | | | | | | | | | | | |
| Maximum | 437.3 | 437.7 | 438.9 | 439.3 | 444.6 | 443.5 | 60.1 | 60.1 | 60.1 | 60.1 | 60.2 | 60.5 |
| Minimum | 432.8 | 433.2 | 434.4 | 434.8 | 426.4 | 433.0 | 59.9 | 59.9 | 59.9 | 59.9 | 59.7 | 59.9 |
| Average | 435.5 | 435.7 | 437.1 | 437.3 | 434.8 | 436.7 | 60.0 | 60.0 | 60.0 | 60.0 | 59.9 | 60.0 |
| Maximum rate of change (%) | 0.99 | 0.99 | 1.00 | 1.00 | 1.01 | 1.01 | 1.00 | 1.00 | 1.00 | 1.00 | 1.00 | 1.01 |
| Minimum rate of change (%) | 0.98 | 0.98 | 0.99 | 0.99 | 0.97 | 0.98 | 1.00 | 1.00 | 1.00 | 1.00 | 1.00 | 1.00 |
| Regulation (%) | ±0.99 | ±0.99 | ±0.99 | ±0.99 | ±0.99 | ±0.99 | ±1.00 | ±1.00 | ±1.00 | ±1.00 | ±1.00 | ±1.00 |

**Table 18.** Power data analysis before and after diesel generators outage.

| Item | Voltage Fluctuation | | | | | | Frequency Fluctuation | | | | | |
| | MCFC | | ESS | | D/G | | MCFC | | ESS | | D/G | |
| | Bef | Aft | Bef | Aft | Bef | Aft | Bef | Aft | Bef | Aft | Bef | Aft |
|---|---|---|---|---|---|---|---|---|---|---|---|---|
| Condition of Synchronization | | | | | | | | | | | | |
| Maximum | 439.1 | 439.7 | 440.7 | 441.2 | 439.0 | 444.8 | 60.1 | 60.1 | 60.1 | 60.1 | 60.1 | 60.3 |
| Minimum | 435.3 | 436.1 | 436.9 | 437.7 | 435.2 | 436.1 | 60.0 | 59.9 | 60.0 | 59.9 | 60.0 | 59.9 |
| Average | 437.4 | 437.7 | 439.0 | 439.3 | 437.3 | 439.3 | 60.0 | 60.0 | 60.0 | 60.0 | 60.0 | 60.0 |
| Maximum rate of change (%) | 1.00 | 1.00 | 1.00 | 1.00 | 1.00 | 1.01 | 1.00 | 1.00 | 1.00 | 1.00 | 1.00 | 1.01 |
| Minimum rate of change (%) | 0.99 | 0.99 | 0.99 | 0.99 | 0.99 | 0.99 | 1.00 | 1.00 | 1.00 | 1.00 | 1.00 | 1.00 |
| Regulation (%) | ±0.99 | ±0.99 | ±1.00 | ±1.00 | ±0.99 | ±1.00 | ±1.00 | ±1.00 | ±1.00 | ±1.00 | ±1.00 | ±1.00 |

Since when synchronized to ESS and diesel generators, voltage stability was confirmed with the voltage fluctuation rate measured between ± 1.01 % (standard range of +6 to −10%).The frequency variation was measured within ± 1.01 %—therefore, it can be seen that frequency fluctuation hardly occurs during system synchronization.

During the breakaways of ESS and diesel generator, it was confirmed that the voltage and frequency were not subjected to the deviation from the acceptable ranges.

## 8. Conclusions

Until now, most of the marine fuel cell research works have been limited to concentrating on the low power capacity of PEMFCs for small vessels or MCFCs for medium and large-sized vessels as auxiliary power sources.

However, in this paper, a hybrid power system was constructed along with a battery and a generator system, in order to apply the MCFC as the main power source for medium and large-sized ships. Their performance in parallel operation was monitored and investigated based on the actual experiment with the test bed.

Experiment results revealed that the deviation levels of voltage and frequency were kept within the standard ranges across all operational cases. Therefore, system synchronization was proven stable. It also confirmed that power quality of the built-in hybrid power source the test bed was compliant with the rules of the Korean Register of Shipping.

Therefore, the novelty of this research can be placed on this contribution to the design and the application of the hybrid power system using MCFC for the propulsion power system of the middle and large-sized ships.

**Author Contributions:** Conceptualization, K.Y. and H.J.; Methodology, K.Y., H.J. and S.K.; Formal analysis, H.J. and S.K.; Resources, H.J. and S.K.; X.X.; Writing—original draft preparation, K.Y. and H.J.; Writing—review and editing, K.Y., H.J. and S.K.; Visualization, H.J. and S.K.; Supervision, K.Y.

**Funding:** This research received no external funding.

**Acknowledgments:** We thank our colleagues from Kido Park, who provided insight and expertise that greatly assisted the research, although they may not agree with all of the conclusions of this paper. The authors would also like to thank POSCO Engineering co. for approval to use 'DFC300' model.

**Conflicts of Interest:** The authors declare no conflict of interest.

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
