# Peer review of "Fuel Cell Application for Investigating the Quality of Electricity from Ship Hybrid Power Sources"

_jmse, doi:10.3390/jmse7080241_

Round 1
Reviewer 1 Report
table 1,2 -- frmat should be checked...text size and design
line 106 -- are located the upper side...should be...are located on the upper side
line 110 -- fuel cell stake...should be...fuel cell stack
Line 95, 120, 140 -- check subsection format
Line 142 -- In addition, the noise is composed of a soundproof type of 75 dB(A) or less....should be ...The system is sound proofed to 75db or less.
Line 181 - 184 -- check format
Line 219 - 224 -- check bullet format
Line 226, 228 -- missing equation number
Line 230 -- check format
Figure 13 quality can be improved
Line 233 and 235 -- units is square brackets??
Table 6 - check format
Results are quite obvious.
Author Response
Dear Sir,
We are very grateful for your line-by-line comments across the manuscript. Your comments are valuable and very helpful for revising and improving our paper, as well as the important guiding significance to our research. We have studied the comments carefully and have revised the manuscript accordingly, which we hope meet with approval.
We put our responses to your comments on the original manuscript as well as the main corrections in the paper and the responses to your official comments are as follows.
All changes were highlighted.
Sincerely yours
Dr. Kyoung Kuk Yoon
Reviewer 1
1. table 1,2 -- format should be checked...text size and design
RESPONSE: Thank you for your detailed feedback. We have modified format following journal guidance and checked all the other tables with instruction.
Table 1. Specification of fuel cell system.
Specification for ‘DFC300MA’. | |
Power Output -Rated output -Voltage -Frequency -Power Quality |
250 kW 380 ~ 480 VAC 50 ~ 60 Hz Per IEEE 519 |
Emissions -NOx -SOx -CO |
<0.02 lb/MWh <0.001 lb/MWh <0.05 lb/MWh |
Table 2. Specification of ESS.
ESS Specification | |
Input Voltage | 440VAC 3phase 3wire(60Hz) |
Output Voltage | 440VAC 3phase 3wire(60Hz) |
Load Capacity | 30 kW |
Battery Ampere-hour | 400Ah |
Inverter Type | Bi-directional grid connected type |
2. line 106 -- are located the upper side...should be...are located on the upper side
RESPONSE:
Thank you for your detailed feedback. As per your opinion, we have modified position of sentence as below.
“The Air injection part and the water discharge part are located on the upper side of MBOP within which two sets of exhaust fans are fitted.”
3. line 110 -- fuel cell stake...should be...fuel cell stack
RESPONSE:
As per your comment, we have changed the word as below.
“Figure 5 describes the concept of the electricity generation process. The system consists of three modes: a 'heat-up mode' for increasing the initial temperature of the fuel cell stack module, a 'ramp-up mode' for increasing the power output to the rated output, and an 'operating mode' for continuing the rated output.”
4. Line 95, 120, 140 -- check subsection format
RESPONSE: Following your suggestion, we have modified below part as journal format.
“3.2.1 MCFC System
The fuel cell to be applied to the test bed is a 300 [kW] class MCFC system ‘DFC300MA’ model manufactured by ‘POSCO Engineering’[28].”
“3.2.2 Energy Storage System (ESS)
ESS refers to a small/medium-sized electrical storage facility that is to store electrical energy and use it when necessary in aids of a distributed power source in a micro-grid. As shown in Figure 6 ESS comprises the battery and Power Conditioning System (PCS).”
“3.2.3 Diesel Generator System
A 50 [kW] diesel generator used in the test bed is a revolving-field type using a permanent magnet. This system is sound proofed to 75db or less. As shown in Figure 7, Tables 3 and 4, it has the synchronous speed of 1800 rpm with four poles.”
5. Line 142 -- In addition, the noise is composed of a soundproof type of 75 dB(A) or less....should be ...The system is sound proofed to 75db or less.
RESPONSE:
As per your opinion, we have modified sentence as below.
“A 50 [kW] diesel generator used in the test bed is a revolving-field type using a permanent magnet. This system is sound proofed to 75db or less.”
6. Line 181 - 184 -- check format
RESPONSE:
Thank you for your detailed feedback. As per your opinion, we have changed format as below.
“Figure 9 represents the configuration of MCFC system so that we can monitor and analyses frequency stabilization time as well as the voltage fluctuation rate and frequency fluctuation rate which can occur in the process of synchronizing the other power sources such as ESS and diesel generator.”
7. Line 219 - 224 -- check bullet format
RESPONSE:
Thank you for your opinion and checked the sentence. But there is no problem with following journal format.
8. Line 226, 228 -- missing equation number
RESPONSE:
According to your instruction, I have inserted number on equation.
“For cargo ships with the main engine power of 10000 kW or above, is defined as equation (1);
(1)
For cargo ships with the main engine power below 10000 kW, is defined as equation (2);
(2)”
9. Line 230 -- check format
RESPONSE:
Thank you for your correction, we have modified style of letter as below senstence.
“5.2. Development of load scenario using real-time operation load pattern”
10. Figure 13 quality can be improved
RESPONSE:
As per your guidance, we have changed figure 13 with more clarification.
11. Line 233 and 235 -- units is square brackets??
RESPONSE:
Following the instruction of journal, we have eliminated bracket () as below sentence:
“Figure 13 and Table 6 summarize the synchronization and off-gird of the load pattern of the normal seagoing case 1 with the peak load of 127 kW of the combined power source. ESS is configured to discharge after one synchronization at a time when the total load of the complex power source is over 100 kW, and then it is changed to the charging mode by the ESS SOC reference value and the charging is progressed in the state connected to the system.”
12. Table 6 - check format
RESPONSE:
Thank you for your comment, we have modified table as below.
Table 6. Synchronization condition in Normal seagoing case 1.
Status | Power sources |
① ESS sync. | MCFC + ESS |
② ESS Charging (keep sync.) | MCFC + ESS |

Reviewer 2 Report
This manuscript has investigated experimentally a hybrid system of a molten carbonate fuel cell, a battery, and a diesel generator for marine power generation. It documented the major components, and the system design, control, analysis, etc. The results showed that the hybrid system can provide or follow the load requirements pretty nicely. Though not much theoretical analysis or technical development is presented here, this real system integration work may find to be interesting to many readers. Therefore, it is recommended to be accepted in this Journal.
Author Response
Dear Sir,
We are very grateful for your line-by-line comments across the manuscript. Your comments are valuable and very helpful for revising and improving our paper, as well as the important guiding significance to our research.
Sincerely yours
Dr. Kyoung Kuk Yoon